# *Oscillatoria limnetica* Mediated Green Synthesis of Iron Oxide (Fe_2_O_3_) Nanoparticles and Their Diverse In Vitro Bioactivities

**DOI:** 10.3390/molecules28052091

**Published:** 2023-02-23

**Authors:** Muhammad Haris, Namra Fatima, Javed Iqbal, Wadie Chalgham, Abdul Samad Mumtaz, Mohamed A. El-Sheikh, Maryam Tavafoghi

**Affiliations:** 1Department of Plant Sciences, Faculty of Biological Sciences, Quaid-i-Azam University, Islamabad 45320, Pakistan; 2Department of Botany, Bacha Khan University, Charsadda 24420, Pakistan; 3Department of Mechanical and Aerospace Engineering, University of California, Los Angeles, CA 90095, USA; 4Botany and Microbiology Department, College of Science, King Saud University, Riyadh 11451, Saudi Arabia; 5Department of Bioengineering, University of California, Los Angeles, 410 Westwood Plaza, Los Angeles, CA 90095, USA

**Keywords:** *Oscillatoria limnetica*, (Fe_2_O_3_), green synthesis, characterizations, biological applications

## Abstract

Iron oxide nanoparticles (Fe_2_O_3_-NPs) were synthesized using *Oscillatoria limnetica* extract as strong reducing and capping agents. The synthesized iron oxide nanoparticles IONPs were characterized by UV-visible spectroscopy, Fourier transform infrared (FTIR), X-ray diffractive analysis (XRD), scanning electron microscope (SEM), and Energy dispersive X-ray spectroscopy (EDX). IONPs synthesis was confirmed by UV-visible spectroscopy by observing the peak at 471 nm. Furthermore, different in vitro biological assays, which showed important therapeutic potentials, were performed. Antimicrobial assay of biosynthesized IONPs was performed against four different Gram-positive and Gram-negative bacterial strains. *E. coli* was found to be the least suspected strain (MIC: 35 µg/mL), and *B. subtilis* was found to be the most suspected strain (MIC: 14 µg/mL). The maximum antifungal assay was observed for *Aspergillus versicolor* (MIC: 27 µg mL). The cytotoxic assay of IONPs was also studied using a brine shrimp cytotoxicity assay, and LD_50_ value was reported as 47 µg/mL. In toxicological evaluation, IONPs was found to be biologically compatible to human RBCs (IC_50_: >200 µg/mL). The antioxidant assay, DPPH 2,2-diphenyl-1-picrylhydrazyly was recorded at 73% for IONPs. In conclusion, IONPs revealed great biological potential and can be further recommended for in vitro and in vivo therapeutic purposes.

## 1. Introduction

Nanotechnology is advantageous and is a widely growing branch of science that has gained the significant attention of the scientific community in the recent era of modern technology [1]. In the last few decades, nanotechnology has gained more attention due to its distinct size-related effect [2]. Metal nanoparticles range from 1 to 100 nm and show unique and fascinating properties such as physical, chemical, optical scattering, and biological properties [3,4]. Among the different nanoparticles, IONPs have gained more importance due to their multifunctional applications in different fields such as the food industry, biotechnology, tissue engineering, and environmental bioremediations [5,6]. Iron oxide NPs have important applications in the field of science for wastewater treatment, drug delivery, magnetic resonance imaging, etc. [7,8].

The NPs are synthesized by various methods including physical, chemical, and biological methods [9,10]. Nanoparticle synthesis through chemical and physical methods requires expensive equipment, uses toxic chemical substances, and has major environmental effects on both biotic and abiotic components of the ecosystem [11,12]. Nanoparticles possess promising potential and could be defined by their minute size, large surface area, and various shapes [13]. Nanotechnology can produce and manipulate things on the atomic scale ranging from 1 to 100 nm [14]. Nanoparticles are considered more important by virtue of their large surface area to volume ratio, which is why they are used in different fields such as biotechnology, tissue engineering, medicine, cosmetics, engineering, electronics, environmental bioremediation, and materials science [15,16]. The biological method of NP synthesis is preferred over physical and chemical methods, as the biological method is ecofriendly, less expensive, pollutant free, biosafe, and biocompatible [6,17]. In addition, nanoparticles synthesized with the help of physical and chemical methods are not friendly to the environment as they use toxic chemical substances and expensive equipment [18,19,20].

Different species of algae such as green algae, diatoms, and cyanobacteria have also been used as biotemplates for the green synthesis of nanoparticles and are considered very important due to the presence of biologically active chemical compounds and secondary metabolites which function as strong reducing, stabilizing, and capping agents [21,22,23]. Among the different inorganic nanomaterials, IONPs have shown unique and fascinating properties with functional adaptability; these characteristics give them applications in cosmetics products, nonlinear optics, biosensors, fibers, antimicrobials, etc. [16,24,25]. Nanoscale IONPs are attracting the particular attention of the scientific community by virtue of their antimicrobial action against bacteria, fungi, and viruses in contrast with various other nanoparticles [26,27]. Some new studies explained the potential of IONPs for environmental remediation and for their ability to reduce environmental pollution [28,29].

Iron oxide exists in various forms such as magnetite (Fe_3_O_4_), hematite (α-Fe_2_O_3_), and Maghemite (β-Fe_2_O_3_); of these forms_,_ hematite is characterized by many significant properties [30]. Hematite of the n-type exists in different shapes such as wire, plate, and shuttle [31]. This is the first report of the synthesis of iron nanoparticles using algae from Pakistan. The aim of the current study was to establish an innovative protocol for green synthesis of hematite-phase IONPs using the algal extract of *Oscillatoria limnetica* as a strong capping and reducing agent without the addition of different reducing and capping agents as used in chemical approaches. Moreover, different characterization techniques such as UV-spectroscopy Fourier Transform Infrared Spectroscopy (FTIR), X-ray Diffractive Analysis (XRD), Energy Dispersive X-ray analysis (EDX), and Scanning Electron Microscopy (SEM) analysis were used to determine the physical and chemical properties of IONPs. Furthermore, different bioactivities were performed to investigate the biomedical potentials of synthesized IONPs.

## 2. Results and Discussion

### 2.1. Biosynthesis and Characterization of IONPs

Nanoparticle synthesis using algal extract has advantage over physical and chemical methods, since the later involve use of hazardous chemicals and require tedious time-consuming procedures. Earlier studies have shown that hazardous chemicals may get adsorb on the surface of nanoparticles during chemical synthesis, consequently cannot be used for biomedical applications [32]. The biosynthesis of iron NPs was explained for the first time using aqueous algal extract of *Oscillatoria limnetica*. The importance of this genus is well recognized. Recent phytochemical studies showed that algae are a rich source of proteins, carbohydrate, terpenoids and glycosides [33]. These chemicals play a significant role in the reduction, stabilization and capping of nanoparticles. Algae-mediated synthesis of nanoparticles start once the precursor salt FeCl_3_.6H_2_O was added into *Oscillatoria limnetica*-mediated extract. The change in color of the solution at 80 °C showed the formation of iron NPs. The change in color of algal extract is due to surface plasmon vibrations [34]. A precise biosynthesis mechanism is shown in Figure 1.

### 2.2. UV-Visible Spectroscopy

IONP synthesis in aqueous solution was further confirmed by UV-visible spectroscopy, scanned at 350–600 nm. The highest absorbance peak was found at 471 nm. This highest absorbance peak revealed the synthesis of IONPs, which falls in the range of surface plasmon resonance of IONPs as shown in Figure 2. UV analysis plays a vital part in the characterization of iron NPs and can be used to obtain important information with regards to shape and size in addition to the stability of IONPs [35]. The results are matched with a previous report [7].

### 2.3. Fourier Transform Infrared Spectroscopy

The oscillation properties of biosynthesized iron NPs were evaluated using FTIR spectral analysis in the range of 500–4000 cm^−1^. The results showed parallel regions for IR absorption. FTIR analysis plays a vital role in examining the functional groups of many organic and inorganic compounds. FTIR spectra show medium sharp peaks at 3781 cm^−1^. The sharp bond at 3781 cm^−1^ is linked to the stretching vibration of the O-H bond. The peak at 3348 cm^−1^ depicted the N-H stretching of amine. The peak at 2326 cm^−1^ signified the strong bond stretching of carbon dioxide (O=C=O). The peak at 1629 cm^−1^ was linked to weak bond stretching of alkene (C=C). The peak at 1440 cm^−1^ corresponded to the methyl group, with a medium bond stretching of alkane (C-H). Furthermore, the peak at 1010 cm^−1^ corresponded to carboxylic acid, O-H stretching. The peaks lower than 1000 cm^−1^ showed a strong C-I stretching with a halo groups, depicted different compounds adsorbed on the surface of nanoparticles, consequent to different functional groups present in *Oscillatoria limnetic* which stabilized the IONPs (Figure 3). The FTIR analysis results are consistent with an earlier study [36].

### 2.4. X-ray Diffractive Analysis

X-ray powder diffraction (XRD) XRD spectrum has confirmed the single phase and formation of crystalline nature of iron oxide NPs. X-ray diffractive analysis has confirmed the development of iron NPs. The XRD spectrum explained that particles are in crystalline nature. The analyzed results showed that the shape of the crystalline was trigonal rhombohedral, and the size of the iron nanoparticles was obtained at a range of 23.33 nm using Debye Scherer equation. The 2Ɵ size of the XRD pattern was in the range from 10° to 80°. There are different peaks of iron nanoparticles at 6.24°, 9.99°, 20.49°, 22.39°, 31.69°, 32.22°, 35.72°, 54.19°, 55.22°, 56.47°, and 75.14° which are Miller indexed to 100, 131, 400, 302, 123, 512, 110, 804, 530, 311, and 372. Bragg reflection of rhombohedral crystalline phase of iron NPs are shown in Figure 4. Some other peaks were also present in algal extract due to stabilizing agents as protein and enzyme [37]. The distinct analysis of XRD confirms the trigonal rhombohedral morphology of IONPs, which is confirmed through JCPD card No. 96-101-1241. The mean crystal size of IONPs was determined through different peaks from FWHMs, the average crystal size of iron nanoparticles was 23.33 nm, according to Scherer’s equation as given, D = K ƛ/β_1/2_CosƟ. The results agreed with a previous report of [38].

### 2.5. Scanning Electron Microscopy

The morphology of biosynthesized iron NPs was confirmed by scanning electron microscopy. Scanning electron microscopy was performed to identify the shape of biosynthesized IONPs (Figure 5), which confirm the formation of trigonal rhombohedral crystalline shapes in line with the XRD data. Large particles have gained specific shape due to crystal growth. SEM image of Fe_2_O_3_-NPs showed that these NPs were present in contact with each other due to magnetic properties of IONPs [39,40]. Iron nanoparticles were characterized by SEM to ascertain the size and physical dimensions of the nanoparticles [41].

### 2.6. Energy Dispersive X-ray Analysis (EDX) Analysis

The elemental composition analysis showed that iron and oxygen were present in all samples. The occurrence of Iron (Fe) in elemental form was confirmed by energy-dispersive analysis X-ray spectra as depicted in the absorption peak visible in the range of 6–7 keV (Figure 6). The EDX spectrum was obtained for the elemental composition which was present in iron nanoparticles. The occurrence of high peak of (Cl) and (O) show that the iron nanoparticle powder is in the chloride and oxide form. Due to surface plasmon resonance of iron nanoparticles, the absorbance peaks were present between 6 and 7 KeV. However, some other peaks and additional elements were also observed, namely Chlorine, Calcium, Oxygen, Sodium and Sulphur. Among other elements the occurrence of protein was also indicated [42]. However, in EDX spectra, (Fe) elements were observed with the highest percentage, which suggests that the major part was iron nanoparticles (Fe_2_O_3_). Hence, EDX analysis produced the qualitative and quantitative status of the Fe elements involved in the formation of IONPs [43].

### 2.7. Antibacterial Assay

Antibacterial assay against different Gram-positive and Gram-negative strains were evaluated using the disc diffusion method. The Gram-positive strains that were utilized in current study were *S. aureus* and *B. subtilis,* whereas the Gram-negative strains were *E. coli* and *P. aeruginosa*. The antimicrobial mechanism of metal NPs against algae pathogens has been rarely studied [8], but the precise mechanism of action of NPs against microbes is not clear; however, different mechanism is involved. The antimicrobial assays of iron NPs are based on a loss of replication that disables the cellular protein and enzyme pathogen [44]. Different studies explain that NPs penetrate the cell membrane and cell wall and disrupt the cell integrity [45]. Some studies suggest NPs induced damage to protein, DNA, and RNA and finally cause cell death [46]. The most infectious diseases are due to bacteria that affect not only the mortality rate of the disease but also the costs of treatment [47]. More use of antibiotics causes different bacterial resistance problems, so scientists are working hard to develop new techniques to reduce the bacterial infections [48]. An antibacterial assay was performed for the following concentrations: 1 = 25 µg/mL, 2 = 50 µg/mL, 3 = 75 µg/mL, 4 = 100 µg/mL, 5 = 125 µg/mL, 6 = 150 µg/mL, and P = Ampicillin at 5 mg/mL. Our results reported that the antibacterial potential increased with increases in concentrations of NPs. Different strains were found to be susceptible to these nanoparticles. *P. aeruginosa* and *B. subtilis* were found to be more susceptible with MIC values of 10.7–14.4 µg/mL. *E. coli* and *S. aureus* were found to be the least effective with MIC values of 35–20 µg/mL. MIC values and ZOI are given in Table 1. In this study, the most successful antimicrobial assay was achieved at a higher concentration at 150 µg/mL of IONPs, so it explains why effective antimicrobial activity was achieved at higher concentrations. The maximum inhibition of *S. aureus* had a range of 70 ± 0.03 at 150 µg/mL, *B. subtilis* had a range of 64.4 ± 0.03 at 150 µg/mL, *E. coli* had a range of 75 ± 0.03 at 150 µg/mL, and *P. aeruginosa* had a range of 42 ± 0.03 at 150 µg/mL. Different studies have been published to investigate the biogenic potential of nanoparticles [49]. An amount of 5 mg of the Ampicillin drug was taken as a positive control. No single concentration was found to be more effective than positive control Ampicillin. The antibacterial assay against different concentrations is shown in Figure 7. The results were confirmed by the previous report of [50].

### 2.8. Antifungal Assay

To study the antifungal potential of biosynthesized IONPs, 30 mg of iron nanoparticles were dispersed in 30 mL of Dimethyl sulfoxide (DMSO) [41]. Various concentrations of IONPs and chemical fungicide (Fluconazole) showed variable growth inhibition (Figure 8 and Table 2). Antifungal assays of biosynthesized IONPs were evaluated against various fungal strains in the concentration range of 50–200 µg/mL. The different fungal strains used were *Rhizopus microsporus* and *Aspergillus versicolor*. Previously, iron NPs were shown to arrest the mycelial growth of *Rhizopus microsporus* and *Aspergillus versicolor* [51]. A recent report on the molecular level of *Rhizopus microsporus* and *Aspergillus versicolor* in response to iron NPs showed the generation of reactive oxygen species (ROS) [51]. Furthermore, the application of iron NPs on tomato seedling has been shown to stimulate the antioxidant potential in hydroponics, which are considered to possibly increase the antimicrobial action of nanoparticles [27]. Increasing concentrations of iron nanoparticles showed positive effects on the growth of these two mycelia, and maximum % inhibition was observed at 200 µg/mL of IONPs. Our results have shown that as we increase the concentration of IONP, there is an increase in the % inhibition and zone of inhibition. Formation of the zone around the well is directly proportional to NP concentrations. The maximum % inhibition of *Rhizopus microsporus* ranged from 47 ± 0.03 to 200 µg/mL, 40 ± 0.03 to 150 µg/mL, and *Aspergillus versicolor* ranged from 73 ± 650.03 to 200 µg/mL, 61 ± 0.03 to (150 µg/mL). *Aspergillus versicolor* was found to be maximally susceptible with an MIC value of (27 µg/mL). *Rhizopus microsporus* was found to be least susceptible with an MIC value of (53 µg/mL). Our study explains that iron NPs have a great potential effect on the formation of spore-producing fungi. The results are consistent with previous study of [52].

### 2.9. Antioxidant Assay

Biosynthesized IONPs have strong potential against antioxidant assay. There are different methods to measure antioxidant assay, but DPPH is most important assay for checking antioxidant potentials of nanoparticles. Antioxidant activities of iron nanoparticles were studied via DPPH assay. DPPH (1,1-diphenyl-2-pyridyl-hydrazine) free radicals were used to test the sample at different concentrations to determine their antioxidant potential [53]. The antioxidant activity of IONPs was studied at different concentrations ranging from 25 to 150 µg/mL. The maximum value of the antioxidant was 73 ± 0.03% at (150 µg/mL). The IONPs scavenged the DPPH free radical at different values of up to 73 ± 0.03, 59 ± 0, 51 ± 0.03, 39.8 ± 0.03, 41 ± 0.002, and 24 ± 0.003% at concentrations of 150, 125, 100, 75, 50, and 25 µg/mL, respectively, as shown in Table 3. The antioxidant assay of IONPs is shown in Figure 9. The results explain that the higher the concentration, the higher the antioxidant activity will be. The results of the antioxidant activity decrease with a decrease in the concentrations of nanoparticles. Standard ascorbic acid was taken as a positive control. The IC_50_ values of all concentrations are shown in Table 3. The results are confirmed by a recent study by [54].

### 2.10. Hemolytic Assay

#### Biocompatibility against Human RBCs

The toxicological and biocompatible nature of iron nanoparticles was investigated against human RBCs, as shown in Figure 10. According to the American Society for materials and testing designation, the biological molecules which show hemolysis below <2% are non-hemolytic, substances which show hemolysis at 2–5% are lightly hemolytic, and substances which show hemolysis at greater than (>5%) are hemolytic [55]. To determine the degree of toxicity of human RBCs, hemolytic assay of iron nanoparticles was studied. The biocompatibility nature of iron nanoparticles was evaluated towards human RBCs by using hemolytic activity in the concentration range of 50 to 200 µg/mL. The concentration of 50 µg/mL showed 7.1% hemolysis, concentration of 100 µg/mL showed 11.3% hemolysis, and concentration of 150 µg/mL showed 14.8% hemolysis. At a concentration of 200 µg/mL, the percentage of hemolysis is 21.5%, that is, it shows strong hemolysis potential. A higher concentration of the hemolytic assay was found in a higher concentration of 200 µg/mL than in lower concentrations, which explains why IONPs released hemoglobin into blood plasma when they contacted the surface of RBCs cells [56]. The biocompatibility nature of iron nanoparticles was assessed against human RBCs from the group O^+^. The maximum and minimum hemolytic potential is shown in Table 4. PBS (phosphate buffer saline) was taken as a negative control. The NPs are considered hemolytic when they ruptured the RBCs, thus releasing hemoglobin. Concentrations lower than 50 µg/mL could be non-hemolytic, so IONPs at lower concentrations against human RBCs confirm their biocompatibility and nontoxic nature. Thus, the results prove that higher concentrations of IONPs will lead to adverse health effects. The results agree with an earlier report [50].

### 2.11. Brine Shrimp Cytotoxicity Assay

Brine shrimp cytotoxicity (BSC) was ascertained to check the cytotoxic potential of IONPs against newly hatched eggs of *A. salina.* BSC is the most suitable test for confirming the potential of biological molecules. The cytotoxic assay of iron nanoparticles was established using *Artemia salina* [57]. It has been explained that the initial developmental stages of *Artemia* are greatly affected by toxins [58]. In the cytotoxicity assay, *Artemia salina* when tested with iron nanoparticles showed the best results as compared to algal extract. The brine shrimp cytotoxicity test is the most favorable cytotoxic screening test for confirming the potential of biological compounds. IONPs examined at four different concentrations (50 µg/mL, 100 µg/mL, 150 µg/mL, and 200 µg/mL) showed % mortality of IONPs at various concentrations (Figure 11). Maximum percentage mortality was found at 200 µg/mL concentrations of iron nanoparticles. The median lethality dose LD_50_ was calculated as 47 µg/mL. Among various concentrations, 50 µg/mL showed the best result with the lowest LD_50_ value. Hence, the BSC assay of IONPs showed a dose-dependent response. Our study explained that % mortality increased with increase in concentration of NPs. The highest concentration was more lethal compared to the lowest concentration [58]. The results are confirmed by a previous study [50].

## 3. Materials and Methods

### 3.1. Chemicals

Various chemicals and catalysts that were used for this research were of scientific grade and were obtained from different chemical suppliers.

### 3.2. Collection and Preparation of Algal Extract

The blue-green algae *Oscillatoria limnetica* was collected from freshwater substrate of the district Mianwali Punjab, Pakistan, in March 2022. The algal sample was brought to the Algal Molecular Genetics Lab of Quaid-i-Azam University, Islamabad, Pakistan. The sample was streaked on the BG11 medium and placed in the growth chamber under yellow light. The authentic literature and a microscope were used for the identification of isolated strains [59,60]. After identification, the material was shade dried and then ground to obtain a fine powder. To obtain algal extract, 1 g of algal powder was mixed with 1 L of deionized water and boiled at 100 °C for 24 h [36]. Furthermore, Whatman filter paper was used to filter the extract and was used for synthesis of NPs. The overall study outline has been summarized in Figure 12.

### 3.3. Synthesis of IONPs

A standard protocol that was established by a previous study for the biosynthesis of IONPs was used with slight modifications [61]. To achieve this purpose, 1 Mm solution of Iron Chloride Hexahydrate (FeCl_3_.6H_2_O) salt (Alfa Aesar, Haverhill, MA, USA) was added to algal extract, and 0.5 L of distilled water was mixed with 0.5 g of algal extract to synthesize iron NPs. On a hot plate, the mixed solution was kept with persistent stirring for 2 h at 80 °C. Initially, the formation of IONPs was determined by a change in color from light brown to dark brown. The solution was subjected to centrifugation at 12,000 rpm for 15 min to obtain pure IONPs. The supernatant was discarded, and the pellet was collected and then wiped with distilled water, followed by drying in an oven at 60 °C for 2 h. Furthermore, calcination was performed in an open-air furnace. Finally, dried IONPs were ground and stored at room temperature for further characterization to confirm their chemical composition and morphology.

### 3.4. Characterization of (IONPs)

#### 3.4.1. UV-Visible Spectroscopy

Iron nanoparticle synthesis was performed by UV spectroscopy (UV-Vis 4000 spectrophotometer, Munich, Germany). The absorbance of iron NPs was evaluated at a wavelength ranging from 350 to 600 nm, and the wavelength of the peak was studied.

#### 3.4.2. Fourier Transform Infrared Spectroscopy

The oscillatory properties were assessed using FTIR spectroscopy. The functional groups involved in stabilization and capping were identified using FTIR (Germany, Starnberg, Perkin Elmer Spectrum 65). The biosynthesized IONPs were coated onto KBr crystal wafers and then drained prior to measurements. The spectral range used to scan the sample was evaluated between 4000 and 500 cm^−1^ and assigned peak numbers.

#### 3.4.3. X-ray Diffraction Analysis

The crystalline nature, size, and phase of IONPs were assessed using XRD. To study crystallographic description of purified IONPs, XRD pattern was evaluated using X-ray diffractive analysis (PANalytical, Eindhoven, The Netherlands) at a 45 kV voltage at 40 mA current. The sample was annealed Copper Kα radiations with a silver monochromator between the 2Ɵ range of 10–80°. Finally, the Debye–Scherer equation, D = K ƛ/β_1/2_CosƟ, was used to calculate size of NPs.

#### 3.4.4. Scanning Electron Microscopy

Scanning electron microscopy was used to assess the morphology and particle size distribution of the sample. The morphological characters and size of IONPs were assessed using SEM (JSM5910, JE0L Tokyo, Japan) with voltages of 1 kV and 5 kV equipped with EDX detector.

#### 3.4.5. Energy Dispersive X-ray Analysis

The composition of elements was studied using EDX. Fundamental analysis of iron nanoparticles was analyzed by an energy dispersive X-ray (EDX) detector.

### 3.5. Antibacterial Assay

Antibacterial assay of *Oscillatoria limnetica*-synthesized IONPs was determined through disc diffusion method [62] against different Gram-positive (*B. subtilis*, *S. aureus*) and Gram-negative (*E. coli*, *P. aeruginosa*) bacterial strains. Pure cultures of bacteria were subculture on nutrient agar media. Furthermore, each strain was washed onto different individual plates using sterile cotton swabs. Whatman filter paper was disinfected by autoclaving. For this purpose, 100 µL of bacterial strain was used to obtained bacterial lawns. After this, filter discs (5 mm) filled with 10 µL of test sample were loaded on bacterial lawn. In total, 10 µL of ampicillin disc was used as a positive control. Iron nanoparticle solution was loaded on each disc and were allowed to dry. Then, these dry discs were placed on inoculated agar media. Later, these petri plates were incubated at 37 °C. After 24 h, the zone of inhibition was measured via vernier caliper. MIC values were determined in the concentration range of 150–25 µg/mL. Various stages of percentage inhibition of bacteria were measured by using the formula given below.
% Inhibition = [1 − (Sample absorbance)/(Absorbance of control)] × 100

### 3.6. Antifungal Assay

The antifungal assay of IONPs was performed using different fungal strains (*Rhizopus microsporus* and *Aspergillus versicolor*). Preserved fungal cultures were refined on PDA media at 26 °C for 7 days. The antifungal activity of iron nanoparticles was ascertained with the help of the poisoned food technique [63]. The PDA media was mixed with different concentration of IONPs in the range of 200–50 mg/mL. In the center of nanoparticle-amended PDA plates, cork borer was used to insert the 4 mm inoculum disc of the fungal strains. Petri plates were incubated at 26 °C for 7 days. A chemical fungicide, Fluconazole, was used as a positive control. Percentage inhibition was measured according to the following formula.
% Growth inhibition = [(Control − Treated)/(Control)] × 100

### 3.7. Antioxidant Assay

The antioxidant assay of iron nanoparticles was performed in terms of free radical activity using the DPPH assay [64]. The free radical scavenging assay was evaluated using different concentrations of IONPs ranging from 150 to 25 µg/mL. The DPPH solution was prepared at room temperature with 4 mg of 0.02 mM DPPH mixed in 100 mL of methanol. Ascorbic acid was taken as positive control, and DMSO was taken as a negative control. Later, incubation was performed for 30 min, and readings were noted at 517 nm to evaluate the percent scavenging of DPPH using the following formula.
% FRSA = [(Control Absorbance − Sample absorbance)/(Control absorbance)] × 100

IC_50_ was determined using a linear regression curve.

### 3.8. Hemolytic Assay

#### Biocompatibility against Human (RBCs)

Hemolytic activity was ascertained to evaluate the biocompatible nature of iron NPs with the aid of freshly extracted human RBCs [65]. To achieve this purpose, 2 mL of fresh human blood was collected and centrifuged at 15,000 rpm for 10 min. After this, plasma was removed, and 5 mL of 7.4 pH Phosphate Buffered Saline (PBS) was added, and the mixture was again centrifuged at 14,000 rpm for 5 min to remove the PBS residue. Then, 100 µL of blood was added to different concentrations of IONPs followed by incubation for 1 h at 35 °C, followed by centrifugation at 12,000 rpm for 15 min. The supernatant was transferred to a 96-well plate, and the reading was recorded at 530 nm to find the percent hemoglobin released. The PBS was used as a negative control, and Triton X-100 was used as the positive control. % Hemolysis was determined as follows.
% Hemolysis = [(Absorbance sample) − (Absorbance of negative control/(Absorbance of positive control)] × 100

### 3.9. Cytotoxicity Assay

#### Brine Shrimp Cytotoxicity Assay

Larvae of brine shrimp *Artemia salina* were used for the cytotoxicity assay to ascertain the in vitro cytotoxicity potential of IONPs [19]. *Artemia salina* eggs were incubated for 24–48 h under light at 30 °C in 1 L of sterile sea saltwater in a glass jar with continuous aeration. Once the larvae were hatched, active free floating nauplii were collected under light conditions and used for further analysis. Subsequently, 0.5 mL of iron NPs with different concentrations was transferred to the nauplii in each well. Vincristine sulphate was taken as the positive control, whereas DMSO was taken as the negative control. Under light conditions, each nauplii was transferred to different concentrations of iron NPs, respectively. The percentage of dead shrimps was determined in each well after incubation for 24 h, and the median lethality dose (LD_50_) was calculated using GraphPad software.

## 4. Conclusions

Successful biosynthesis of crystalline hematite phase iron NPs has been synthesized using a novel, ecofriendly, and green synthesis protocol. Synthesized IONPs were extensively characterized using different techniques, such as UV, FTIR, XRD, SEM, and EDX. Our results explain effective biological properties of the iron NPs. Biosynthesized IONPs indicated potential in vitro biological activities such as antibacterial and antifungal activities. Furthermore, a moderate DPPH antioxidant assay was performed, and IONPs were found to be biocompatible using erythrocytes. Biosynthesized iron NPs were found to be minimally toxic to normal human RBCs. This study also hypothesized that the type of algal material can have important effects on its biomedical applications. In conclusion, our results showed that IONPs can be designed for different treatments. Moreover, we suggest further studies on the toxicity and biocompatibility aspects to further reveal their biosafe and biocompatible nature.

## Figures and Tables

**Figure 1 molecules-28-02091-f001:**
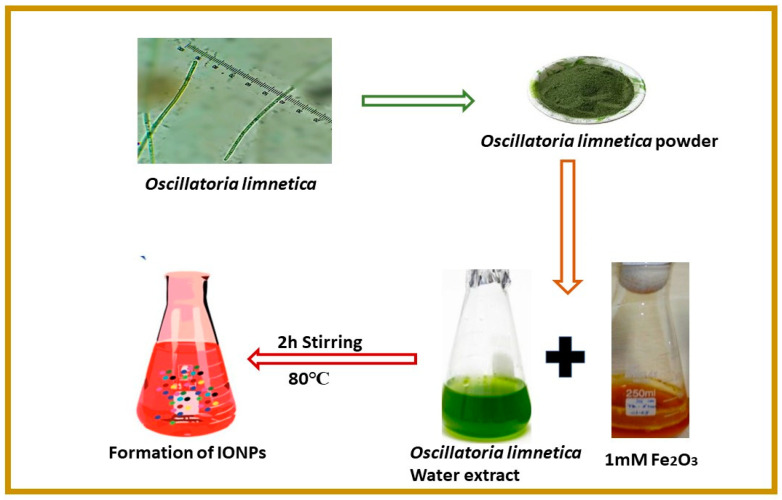
Novel protocol showing green synthesis of *Oscillatoria limnetica*-mediated IONPs.

**Figure 2 molecules-28-02091-f002:**
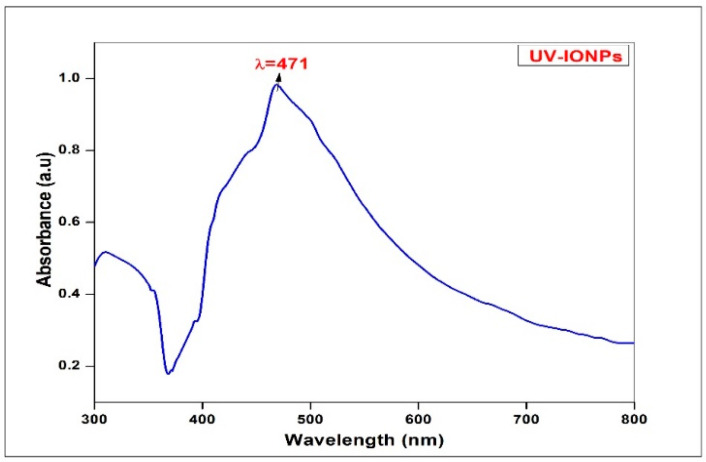
UV-visible spectra of *Oscillatoria limnetica*-mediated IONPs.

**Figure 3 molecules-28-02091-f003:**
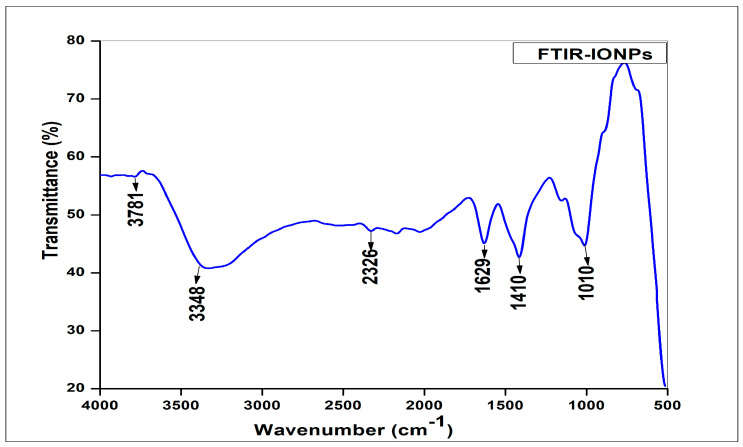
FTIR spectra showing different functional groups involved in the biosynthesis of IONPs.

**Figure 4 molecules-28-02091-f004:**
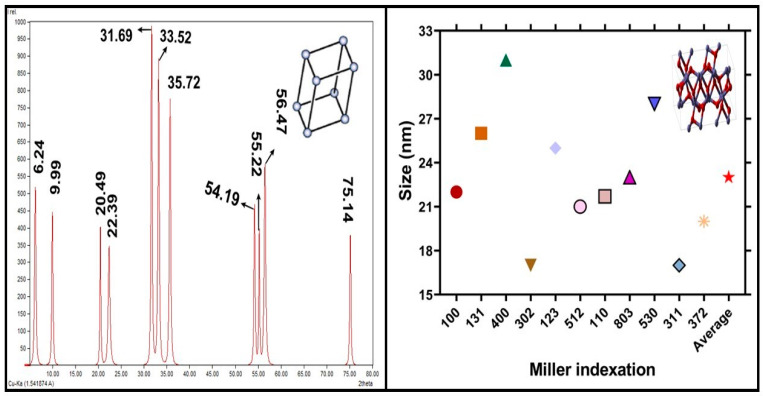
X-ray diffractive analysis of IONPs NPs. Size calculation via Scherer equation.

**Figure 5 molecules-28-02091-f005:**
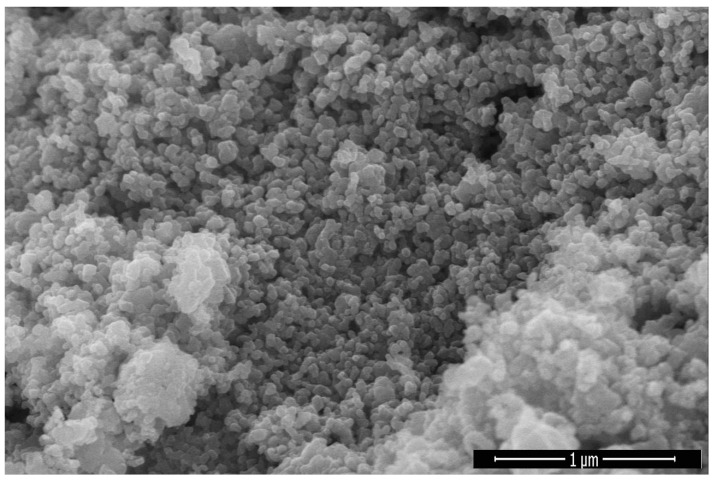
SEM image of *Oscillatoria limnetica*-mediated IONPs.

**Figure 6 molecules-28-02091-f006:**
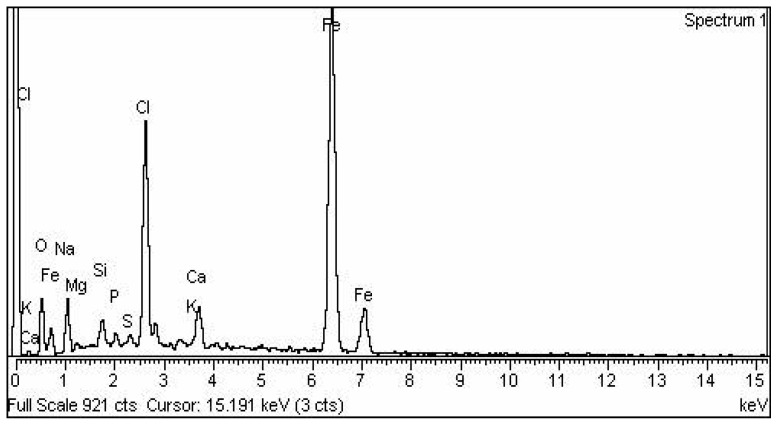
Energy-dispersive X-ray spectroscopy (EDX)EDX spectra showing the elemental composition of IONPs.

**Figure 7 molecules-28-02091-f007:**
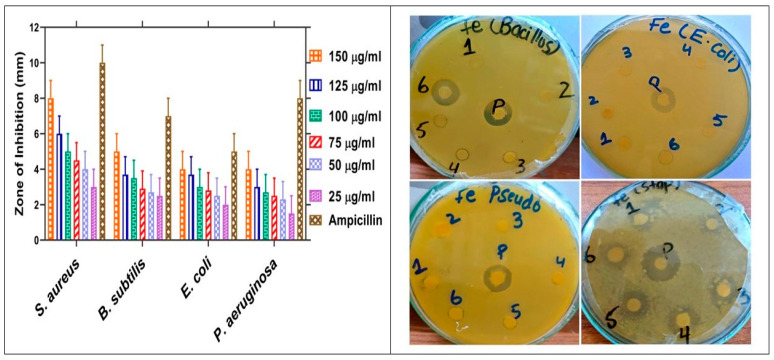
Antibacterial potential of *Oscillatoria limnetica*-mediated IONPs.

**Figure 8 molecules-28-02091-f008:**
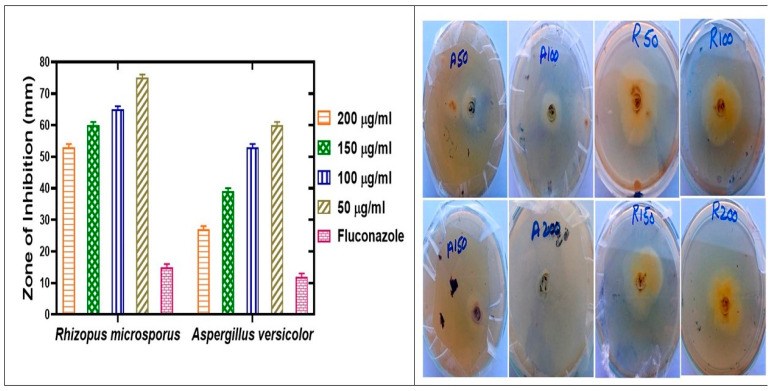
Antifungal potential of *Oscillatoria limnetica*-mediated IONPs.

**Figure 9 molecules-28-02091-f009:**
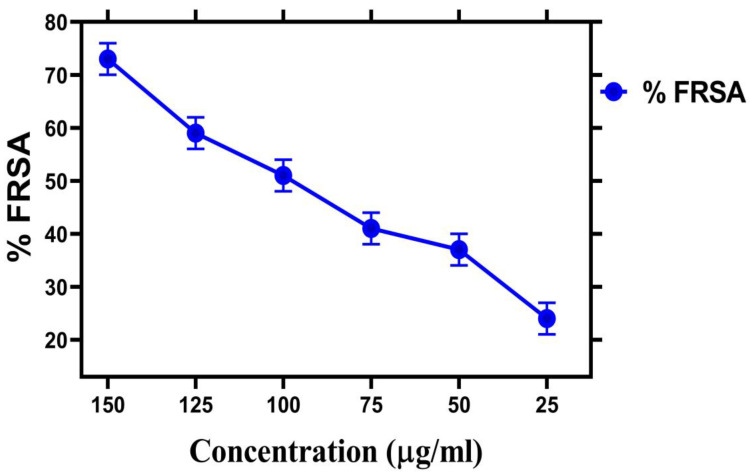
Antioxidant potential of a synthesized IONPs.

**Figure 10 molecules-28-02091-f010:**
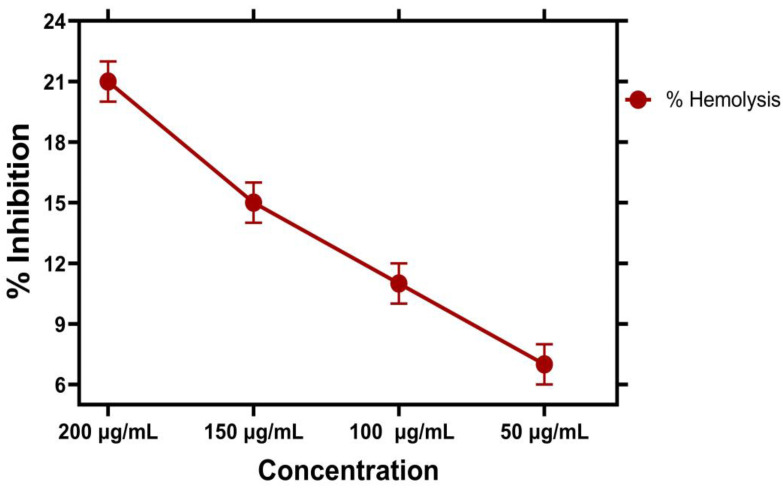
Biocompatibility potential of IONPs against human red blood cells RBC.

**Figure 11 molecules-28-02091-f011:**
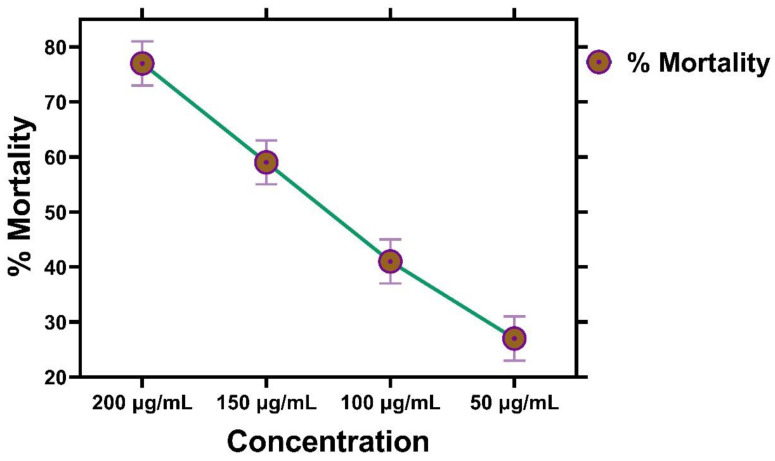
Cytotoxicity potential of IONPs against brine shrimp.

**Figure 12 molecules-28-02091-f012:**
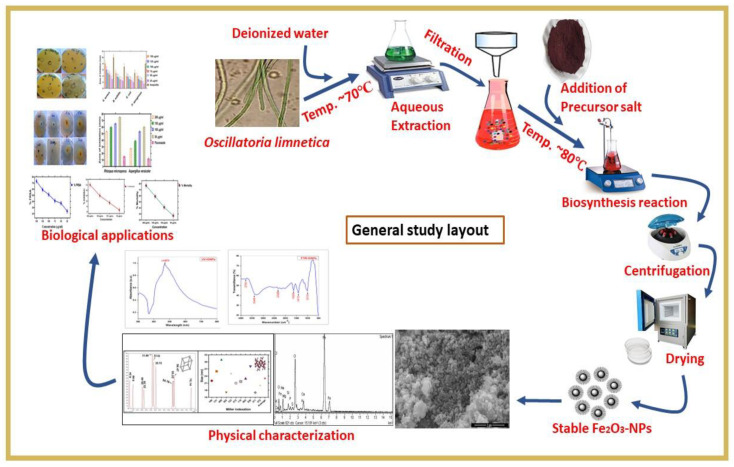
Detailed diagram showing green synthesis, characterization, and diverse biological application of IONPs.

**Table 1 molecules-28-02091-t001:** Zone of inhibition (ZOI) and Minimal inhibitory concentrations (MIC) calculations of different bacterial strains.

Compound	Concentrations	Zone of Inhibition (ZOI) and (% Inhibition)							
		Gram-Positive Bacteria				Gram-Negative Bacteria			
		*S. aureus*		*B. subtilis*		*E. coli*		*P. aeruginosa*	
		ZOI (mm)	(% Inhibition)	ZOI (mm)	(% Inhibition)	ZOI (mm)	(% Inhibition)	ZOI (mm)	(% Inhibition)
IONPs	25 µg/mL	3 ± 0.03	20 ± 0.03	2.5 ± 0.03	14.4 ± 0.03	2 ± 0.03	35 ± 0.03	1.5 ± 0.03	10.7 ± 0.03
	50 µg/mL	4 ± 0.03	30 ± 0.03	2.7 ± 0.03	31.5 ± 0.03	2.5 ± 0.03	45 ± 0.03	2.3 ± 0.03	20.7 ± 0.03
	75 µg/mL	4.5 ± 0.03	35 ± 0.03	2.9 ± 0.03	34.5 ± 0.03	2.8 ± 0.03	51 ± 0.03	2.5 ± 0.03	23.2 ± 0.03
	100 µg/mL	5 ± 0.03	40 ± 0.03	3.5 ± 0.03	43 ± 0.03	3 ± 0.03	55 ± 0.03	2.7 ± 0.03	25.7 ± 0.03
	125 µg/mL	6 ± 0.03	50 ± 0.03	3.7 ± 0.03	45.8 ± 0.03	3.7 ± 0.03	69 ± 0.03	3 ± 0.03	29.5 ± 0.03
	150 µg/mL	8 ± 0.03	70 ± 0.03	5 ± 0.03	64.4 ± 0.03	4 ± 0.03	75 ± 0.03	4 ± 0.03	42 ± 0.03
Ampicillin	5 mg/mL	10 ± 0.05	100 ± 0.05	7 ± 0.05	100 ± 0.05	5 ± 0.05	100 ± 0.05	8 ± 0.05	100 ± 0.05
		Minimal inhibitory concentrations (MIC)							
		MIC (µg/mL)		MIC (µg/mL)		MIC (µg/mL)		MIC (µg/mL)	
IONPs		20 ± 0.03		14.4 ± 0.03		35 ± 0.03		10.7 ± 0.03	

**Table 2 molecules-28-02091-t002:** Zone of inhibition (ZOI) and minimum inhibitory concentration (MIC) ZOI and MIC calculation of fungal strains.

Compound	Concentrations	Zone of Inhibition (ZOI) and (% Inhibition)			
		Fungal Strains			
		*Rhizopus microsporus*		*Aspergillus versicolor*	
		ZOI (mm) and (% Inhibition)		ZOI (mm) and (% Inhibition)	
IONPs	50 µg/mL	75 ± 0.03	25 ± 0.03	60 ± 0.03	40 ± 0.03
	100 µg/mL	65 ± 0.03	35 ± 0.03	53 ± 0.03	47 ± 0.03
	150 µg/mL	60 ± 0.03	40 ± 0.03	39 ± 0.03	61 ± 0.03
	200 µg/mL	53 ± 0.03	47 ± 0.03	27 ± 0.03	73 ± 0.03
(Fluconazole)	5 mg/mL	15 ± 0.05	100 ± 0	12 ± 0.05	100 ± 0
	Minimal inhibitory concentrations (MIC)				
		MIC (µg/mL)		MIC (µg/mL)	
IONPs		53 ± 0.03		27 ± 0.03	

**Table 3 molecules-28-02091-t003:** IC_50_ and FRSA values calculations for antioxidant assay.

Concentration of NPs	% FRSA	IC50
25 µg/mL	24 ± 0.003	1.32
50 µg/mL	37 ± 0.002	3.9
75 µg/mL	41.8 ± 0.03	6.4
100 µg/mL	51 ± 0.03	9.02
125 µg/mL	59 ± 0	11.6
150 µg/mL	73 ± 0.03	14.2

**Table 4 molecules-28-02091-t004:** IC_50_ and hemolysis values for biocompatibility assay.

Concentrations of NPs	% Hemolysis	Triton X-100	IC50
50 µg/mL	7.1 ± 0.003	0.200 ± 0.40	49.5
100 µg/mL	11.3 ± 0.005	0.62 ± 0.90	99.5
150 µg/mL	14.8 ± 0.006	0.79 ± 0.01	149.5
200 µg/mL	21.5 ± 0.003	0.180 ± 0.20	199.5

## Data Availability

All the raw data of this research can be obtained from the corresponding authors upon reasonable request.

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
