# Peer review of "Oscillatoria limnetica Mediated Green Synthesis of Iron Oxide (Fe2O3) Nanoparticles and Their Diverse In Vitro Bioactivities"

_molecules, 2023, doi:10.3390/molecules28052091_

Round 1

Reviewer 1 Report

A green method to synthesize iron oxide nanoparticles was presented in this manuscript. And the in vitro bioactivities of the nanoparticles were investigated.

1.     Please give the full name of IONPs.

2.     There are so many formatting errors, such as

“Iron oxide exist in various shapes like magnetite (Fe2O3) and Hematite (α-Fe2O3) and Maghemite (β-Fe2O3) in entire these forms, hematite grasp much significant properties [15].”

3.     From the SEM images, it is very difficult to distinguish the morphology of the fabricated samples. It is hard to tell that the fabricated samples are nanoparticles. Moreover, it is difficult to make sure the particles size from the SEM analyses.

4.     α-Fe2O3, β-Fe2O3, and γ-Fe2O3 exhibit similar XRD patterns. If possible, please supplement XPS analyses.

5.     What is the mechanism of the in vitro bioactivities of the nanoparticles?

Author Response

Response to Reviewer Comments

Thank you so much for reviewing our manuscript by providing your quality time, valuable comments/ recommendations that has really improved the quality and structure of our titled manuscript. Our team members have extensively reviewed and properly addressed all suggestions/recommendations point by point as the worthy reviewers suggested and now we hope that the revised manuscript is highly improved. We again critically reviewed the manuscript several times for typos errors and other mistakes. If you recommend further suggestions, let us know, we will be very happy to address.

Reviewer 1

Comment: Please give the full name of IONPs.

Response: Thank you so much for your quality time and comprehensive review by highlighting the mistake. Highly appreciated. The full name of IONPs is provided in the revised manuscript and highlighted yellow wherever needed.

       Comment: There are so many formatting errors, such as

“Iron oxide exist in various shapes like magnetite (Fe2O3) and Hematite (α-Fe2O3) and Maghemite (β-Fe2O3).

Response: The formatting errors have been corrected and replaced in the statement provided “Iron oxide exist in various shapes like magnetite (Fe3O4) and Hematite (α-Fe2O3) and Maghemite (β-Fe2O3) and highlighted yellow in the revised manuscript. The authors have once again comprehensively reviewed the whole manuscript for formatting errors and made corrections wherever needed.

Comment: From the SEM images, it is very difficult to distinguish the morphology of the fabricated samples. It is hard to tell that the fabricated samples are nanoparticles.

Response: Respected reviewer thanks for your comments and highlights. We have addressed this comments and provided a new SEM image to the revised manuscript.

Comment: α-Fe2O3, β-Fe2O3, and γ-Fe2O3 exhibit similar XRD patterns. If possible, please supplement XPS analyses.

Response: Dear reviewer, thank you for extensively reviewing our manuscript. Our XRD patterns determined the synthesis of α-Fe2O3 nanoparticles. We appreciate your suggestion, but unfortunately, we don’t have XPS facility here in Pakistan. We have already possible done extensive characterizations which confirm the successful biosynthesis of α-Fe2O3Nanoparticles. Hope you understand our situation and will cooperate in this regard. We will perform XPS in our upcoming manuscripts which are under process.

Comment: What is the mechanism of the in vitro bioactivities of the nanoparticles?

Response: The mechanism of the in vitro bioactivities of the nanoparticles is provided in detail in the results and discussion of the revised manuscript. Changes made have been highlighted yellow.

Reviewer 2 Report

The manuscript entitled “Oscillatoria limnetica mediated Green Synthesis of Iron Oxide Nanoparticles and their Diverse In vitro Bioactivities” aims to establish an innovative protocol for green synthesis of hematite phase IONPs using algal extract of Oscillatoria limnetica as a strong capping and reducing agents without the addition of different reducing and capping agents as used in chemical approaches. Various characterization techniques, such as UV-spectroscopy, FTIR, XRD, EDX, and SEM analysis were used to determine the physical and chemical properties of IONPs. In addition, various bioactivities were performed to investigate the potential biomedical applications of synthesized IONPs. The idea of the manuscript is good, and this topic is important. The manuscript is well prepared in general. The results are well presented, and the discussion is reasonable. The conclusions are in line with the results. Still, there are some inconsistencies to be improved in order manuscript to be published. My concerns are given below.

Line 84: iron oxide is not a salt. What was the precursor for NPs?

Section 2.6. symbols for the elements are redundant.

The synthesis of NPs is not clear. What was the precursor? What is in the NPs? Is it established? The authors mentioned Fe2O3 and Fe3O4. It should be clarified. This section must be improved.

Why is Figure 1 at the end?

Author Response

Reviewer 2

Response to Reviewer Comments

Thank you so much for reviewing our manuscript by providing your quality time, valuable comments/ recommendations that has really improved the quality and structure of our titled manuscript. Our team members have extensively reviewed and properly addressed all suggestions/recommendations point by point as the worthy reviewers suggested and now we hope that the revised manuscript is highly improved. We again critically reviewed the manuscript several times for typos errors and other mistakes. If you recommend further suggestions, let us know, we will be very happy to address.

Comment: Line 84: iron oxide is not a salt. What was the precursor for NPs?

Response: The precursor salt used was Iron Chloride Hexahydrate (FeCl3.6H2O) for the biosynthesis of Iron oxide nanoparticles has been added into the revised manuscript and highlighted yellow.

Comment: Section 2.6. Symbols for the elements are redundant.

The synthesis of NPs is not clear. What was the precursor? What is in the NPs? Is it established? The authors mentioned Fe2O3 and Fe3O4. It should be clarified. This section must be improved.

Response: Section 2.6 only shows the EDX analysis, it represents the element symbols. We have again reviewed section 2.6 but couldn’t found Fe3O4. Thank you for reviewing the manuscript in detail. The authors have not mentioned Fe2O3 and Fe3O4 in section 2.6 rather it is provided in the last paragraph of introduction, which discuss that different forms of the Iron oxide nanoparticles can be formed during synthesis. But our asynthesized nanoparticles are α-Fe2O3 .The precursor salt used during synthesis was Iron Chloride Hexahydrate (FeCl3.6H2O) has now been added to the revised manuscript.  Further, EDX analysis confirm the synthesis of Fe2O3 during synthesis process which is established from the peaks provided, which clearly indicate peaks for Iron and oxygen. We have addressed this comments throughout the manuscript and made changes wherever needed.

Comment: Why is Figure 1 at the end?

Response: Thanks for highlighting this issue and now we added figure number 1 at appropriate position. Your deep review and focused time is highly appreciated.

Round 2

Reviewer 1 Report

This manuscript can be accepted in present form.

Author Response

Thank you so much once again for your comprehensive review and valuable time. Your comments/ suggestions has really improved the structure and quality of our manuscript. We are happy to hear about your decision. 

Reviewer 2 Report

The Section 2.6. contains the symbols and the names of the determined elements. The authors should decide and leave only names, for example. I suggest removing the symbols, but it could be otherwise, also.

Fe3O4 is mentioned in section 2.5. line 142 of the original manuscript and line 147 of the revised manuscript. Please, look carefully and address it.

Author Response

Thank you so much once again for your comprehensive review and valuable time and up to date cooperation in pursuing this manuscript. Your comments/ suggestions has really improved the structure and quality of our manuscript, which we now hope that It will attract wide readership of the scientific community. We all have once again extensively reviewed the manuscript for all types of major and minor mistakes and removed the shortcomings wherever needed. Our research team have reviewed the manuscript several times for English proficiency to further enhance the quality of sentence structure and grammar wherever need. After doing great team efforts now we are very confident and positive that our article is highly improved and ready for publication. If you need further suggestions, we will be very happy to address. All changes made have been highlighted yellow.  

Comment: The Section 2.6 contains the symbols and the names of the determined elements. The authors should decide and leave only names, for example. I suggest removing the symbols, but it could be otherwise, also.

Response: The symbols have been removed from section 2.6 and now only names are included in the revised manuscript. Thanks a lot for your time and sincere efforts.  

Comment: Fe3O4 is mentioned in section 2.5. line 142 of the original manuscript and line 147 of the revised manuscript. Please, look carefully and address it.

Response: Thank you so much for your deep review. Highly appreciated. Our asynthesized nanoparticles is α-Fe2O3. The word “Fe3O4” has been replaced with Fe2O3-NPs in the revised manuscript.